# Performance Optimization of Hybrid Satellite-Terrestrial Relay Network Based on CR-NOMA

**DOI:** 10.3390/s20185177

**Published:** 2020-09-10

**Authors:** Long Zhao, Tao Liang, Kang An

**Affiliations:** 1College of Communications Engineering, Army Engineering University, Nanjing 210007, China; zhaolong2316@163.com; 2Sixty-Third Research Institute, National University of Defense Technology, Nanjing 210007, China; ankang89@nudt.edu.cn

**Keywords:** satellite communication, non-orthogonal multiple access, decode-and-forward, relay selection, antenna selection, outage probability

## Abstract

The non-orthogonal multiple access (NOMA) scheme realizes the transmission of multiple user signals at the same time and frequency resource block through power domain multiplexing, which improves the system transmission rate and user fairness. In this paper, we propose a joint relay-and-antenna selection scheme based on the cognitive radio scenario. This scheme can achieve the maximum communication rate of the secondary user when the primary user maintains the optimal outage performance. In the considered system both terrestrial relays and users are deployed with multi-antenna configurations and the terrestrial relays adopt the decode-and-forward (DF) strategy to achieve communication between satellites and users. Then, we derive the exact outage probability expression of each user in the system and the asymptotic probability expression under high signal-to-noise ratio (SNR). Numeric results demonstrate that increasing the number of relays and antennas on the terrestrial nodes can both improve system outage performance. Moreover, the number of relays imposes a more obvious effect on the achievable system performance.

## 1. Introduction

By introducing relay technology into the satellite system, the hybrid satellite-terrestrial relay network (HSTRN) can significantly enhance signal coverage strength and coverage of the direct transmission link. To date, several works have examined the key performance indicators of HSTRN [1,2,3,4,5,6,7]. The authors in [8] derived the exact expression of the outage probability of HSTRN system and the asymptotic expression under high signal-to-noise ratio by using Meijer-G function, which revealed the diversity order and coding gain of the system. The work in [9] studied HSTRN based on amplify-and-forward (AF) strategy and analyzed the average symbol error rate of M-ary phase shift keying constellation. In [10], the authors studied the HSTRN based on the multi-antenna DF strategy and analyzed the impact of antenna configuration and satellite link interference on cognitive network performance. The research of the above paper starts from different angles and adopts different strategies to significantly improve the spectral efficiency and transmission reliability of HSTRN. However, almost all of these studies use the OMA scheme. The OMA scheme can only serve one user per time slot/frequency. Although the interference between users is effectively avoided, there is also a phenomenon of low spectrum utilization, and it is also unfair to users with poor channel conditions.

The NOMA scheme uses superimposed coding to enable transmission signals of different users to be transmitted in the same channel. At the same time, the receiver uses the serial interference cancellation (SIC) criterion to eliminate interference to achieve signal extraction and decoding [11,12]. In this scheme, the signal source performs power distribution according to the user’s priority and corresponding channel conditions, which greatly meets the communication needs of different users and improves the fairness of the system.

To date, the theoretical research based on the NOMA scheme is mainly concentrated in the cellular mobile communication network. The authors in [13] studied the downlink NOMA system and found that the system outage probability mainly depended on the user’s target rate and power distribution coefficient. In [14], the authors proposed two antenna selection schemes, max–min–max AS (AIA-AS) and max-max-max AS (A^3^-AS), which have higher computational efficiency for the NOMA system with multiple antennas. Moreover, through comparison, it is found that both schemes can significantly improve user performance. Among them, AIA-AS scheme can provide better system fairness, and A^3^ -AS scheme can provide better overall system rate. The work in [15] studied the CR-NOMA system based on energy harvesting assistance and analyzed the impact of the receiver’s imperfect successive interference cancellation on outage behavior and throughput performance. The authors in [16] studied the NOMA system throughput and the optimal position optimization of the UAV in the delay-constrained transmission mode. The work in [17] introduced the NOMA scheme into a multi-beam satellite system and proposed a scheduling strategy to enhance system performance. The authors in [18,19] integrated the NOMA scheme into HSTRN and analyzed the system outage performance based on AF strategy and the effect of parameter configuration on user performance. By applying the NOMA to enhance spectrum efficiency, the authors in [20] investigated a joint beamforming and power allocation scheme in satellite-terrestrial integrated networks. The authors in [21] introduced NOMA into a satellite communication network without terrestrial networks and studied the physical layer security of satellite downlinks. In [22], the authors studied the uplink land satellite mobile communication system based on NOMA and analyzed the outage performance of the system when the satellite receiver adopted continuous interference cancellation or joint decoding. The work in [23] introduced NOMA into HSTRN and analyzed the system outage performance when the near users of the NOMA group were relayed as far users. The authors in [24,25], respectively studied the outage performance of the HSTRN system based on NOMA under the condition of hardware damage, partial relay selection scheme and imperfect CSI.

However, as of now, there have been no reports about the application of MIMO-NOMA to the HSTRN network. To fill this gap, this paper proposes a joint relay-and-antenna selection algorithm based on MIMO-NOMA. Not only can it maximize the system capacity when taking into account the priority of HSTRN users, but it can also greatly reduce the computational complexity of the system. The rest of this article is organized as follows: Related system models are introduced in Section 2. In Section 3, we derive the exact expression of each user’s outage probability and the asymptotic outage probability expression under high signal-to-noise ratio. In Section 4, the theoretical data simulation of computer software is shown. Finally, the fifth part summarizes the full text.

## 2. System Model

The downlink satellite communication system we considered includes satellite S, N relays, 2 users. Each relay is equipped with K antennas. User 1 and User 2 are equipped with M antennas and L antennas, respectively. There is no direct link between the satellite and the users. They can only assist communication through terrestrial relays. The communication link between the satellite and the terrestrial relays are subject to shadowed Rician fading, and the channel coefficient vector is $i_{n}=\left[i_{1},i_{2},\ldots i_{N}\right]$. The communication link between the terrestrial relays and the users are subject to Nakagami-m fading. The channel coefficient between the relay node n and User 1 is K×M matrix $H_{n}$.$h_{nk}=[h_{nk1},h_{nk2},\ldots ,h_{nkM}]$ represents the channel coefficient vector between the k–th antenna of the relay node n and User 1. The channel coefficient between the relay node n and User 2 is K×L matrix $G_{n}$. $g_{nk}=[g_{nk1},g_{nk2},\ldots ,g_{nkL}]$ represents the channel coefficient vector between the k–th antenna of the relay node n and User 2. To enhance the engineering practicality of the model, we assume that User 1 is a fire sensor in a forest area, and its application requirements are low rate and fast service. On the other hand, User 2 is a general user, and its application requirement is to download files, movies and other common Internet services. The system model is shown in Figure 1.

A complete satellite downlink communication process includes two time slots. In the first time slot, the satellite s sends the superimposed signal $x\left(x=\sqrt{\gamma_{1}P_{s}}x_{1}+\sqrt{\gamma_{2}P_{s}}x_{2}\right)$ to the relays, and the signal received by the relay node n can be written as:(1)$$y_{sn}=i_{n}(\sqrt{\gamma_{1}P_{s}}x_{1}+\sqrt{\gamma_{2}P_{s}}x_{2})+n_{r}$$
where $i_{n}$ is the channel coefficient between the satellite and the terrestrial relay n and $n_{r}$ is the additive white Gaussian noise (AWGN) with variance $\delta_{}^{2}$. $P_{s}$ is the signal transmission power of the satellite s. $\gamma_{1}$ and $\gamma_{2}$ are the power distribution coefficients corresponding to the signal $x_{1}$ and the signal $x_{2}$, respectively. Moreover, $\gamma_{1}\;+\;\gamma_{2}\;=\;1$.

According to the serial interference cancellation (SIC) decoding principle, the relay node n decodes the received signal in the order of channel attenuation from large to small. The signal-to-interference-plus-noise ratio (SINR) of the signal $x_{1}$ and the signal $x_{2}$ received at relay node n can be expressed as:(2)$$SINR_{n}^{1}=\frac{\gamma_{1}\rho_{sn}{\left|i_{n}\right|}^{2}}{\gamma_{2}\rho_{sn}{\left|i_{n}\right|}^{2}+1}$$
(3)$$SINR_{n}^{2}=\gamma_{2}\rho_{sn}{\left|i_{n}\right|}^{2}$$
where $\rho_{sn}=P_{s}/\delta_{}^{2}$.

After the relay node n finishes decoding the signal $x_{1}$ and the signal $x_{2}$, it reshapes the transmission signal according to the superposition coding method. In the second time slot, the relay node n broadcasts the superimposed signal to the users, and the signals received by User 1 and User 2 can be expressed as
(4)$$y_{n1}=h_{n}(\sqrt{\gamma_{1}P_{n}}x_{1}+\sqrt{\gamma_{2}P_{n}}x_{2})+n_{1}$$
(5)$$y_{n2}=g_{n}(\sqrt{\gamma_{1}P_{n}}x_{1}+\sqrt{\gamma_{2}P_{n}}x_{2})+n_{2}$$
where $h_{n}$ and $g_{n}$ are the channel coefficients from the relay n to User 1 and User 2, respectively. $n_{1}$ and $n_{2}$ are AWGN with variance $\delta_{}^{2}$. $P_{n}$ is the transmission power of the relay n. The SINR of the signal $x_{1}$ received by User 1 can be expressed as
(6)$$SINR_{1}^{1}=\frac{\gamma_{1}\rho_{nd}{\left|h_{nkm}\right|}^{2}}{\gamma_{2}\rho_{nd}{\left|h_{nkm}\right|}^{2}+1}$$

The SINR of the signal $x_{1}$ and the signal $x_{2}$ received by User 1 can be written as
(7)$$SINR_{2}^{1}=\frac{\gamma_{1}\rho_{nd}{\left|g_{nkl}\right|}^{2}}{\gamma_{2}\rho_{nd}{\left|g_{nkl}\right|}^{2}+1}$$
(8)$$SINR_{2}^{2}=\gamma_{2}\rho_{nd}{\left|g_{nkl}\right|}^{2}$$
where $\rho_{nd}=P_{n}/\delta_{}^{2}$.

Considering that the communication link between the satellite s and the relay node n is subject to shadowed Rician fading, the probability density function (PDF) of ${\left|i_{n}\right|}^{2}$ can be written as
(9)fin2(x)=αin2exp(−βin2x)F11min2;1;δin2x
where $\alpha_{{\left|i_{n}\right|}^{2}}=0.5{\left(2b_{{\left|i_{n}\right|}^{2}}m_{{\left|i_{n}\right|}^{2}}/\left(2b_{{\left|i_{n}\right|}^{2}}m_{{\left|i_{n}\right|}^{2}}+\Omega_{{\left|i_{n}\right|}^{2}}\right)\right)}^{m_{{\left|i_{n}\right|}^{2}}}$,$\beta_{{\left|i_{n}\right|}^{2}}=0.5/b_{{\left|i_{n}\right|}^{2}}$,$\delta_{{\left|i_{n}\right|}^{2}}=0.5\Omega_{{\left|i_{n}\right|}^{2}}/b_{{\left|i_{n}\right|}^{2}}/\left(2b_{{\left|i_{n}\right|}^{2}}m_{{\left|i_{n}\right|}^{2}}+\Omega_{{\left|i_{n}\right|}^{2}}\right)$, $\Omega_{{\left|i_{n}\right|}^{2}}$ and $2b_{{\left|i_{n}\right|}^{2}}$ represent the average power of the direct component and the multipath component, $m_{{\left|i_{n}\right|}^{2}}$ is the fading parameter of the Nakagami-m distribution and F11(a;b;c) represents the confluent hypergeometric function. By consulting the relevant formulas ([26], Equation (9.14.1), Equation (3.381.1)), cumulative distributed function (CDF) of ${\left|i_{n}\right|}^{2}$ can be expressed as
(10)$$F_{{\left|i_{n}\right|}^{2}}(x)=\alpha_{{\left|i_{n}\right|}^{2}}\sum_{k=0}^{\infty }\frac{{\left(m_{{\left|i_{n}\right|}^{2}}\right)}_{k}\delta_{{\left|i_{n}\right|}^{2}}^{k}}{{\left(k!\right)}^{2}\beta_{{\left|i_{n}\right|}^{2}}^{k+1}}\gamma \left(k+1,\beta_{{\left|i_{n}\right|}^{2}}x\right)$$
where $\gamma \left(k+1,\beta_{{\left|i_{n}\right|}^{2}}x\right)$ is the incomplete Gamma function ([26], Equation (8.350.1)).

Due to the variable $h_{nkm}^{}$ with fading parameter $m_{n1}$ and $g_{nkl}^{}$ with parameter $m_{n2}$ subject to Nakagami-m distribution, we can easily obtain that ${\left|h_{nkm}^{}\right|}^{2}$ and ${\left|g_{nkl}^{}\right|}^{2}$ follow Gamma distribution and the CDF of them can be expressed as follows:(11)$$F_{{\left|h_{nkm}^{}\right|}^{2}}(y)=\int_{0}^{y}\frac{m_{n1}^{m_{n1}}y^{m_{n1}-1}}{\Omega_{n1}^{m_{n1}}\Gamma (m_{n1})}e^{-\frac{m_{n1}y}{\Omega_{n1}}}dy=\frac{\gamma \left(m_{n1},ym_{n1}/\Omega_{n1}\right)}{\Gamma (m_{n1})}$$
(12)$$F_{{\left|g_{nkl}^{}\right|}^{2}}(y)=\int_{0}^{y}\frac{m_{n2}^{m_{n2}}y^{m_{n2}-1}}{\Omega_{n2}^{m_{n2}}\Gamma (m_{n2})}e^{-\frac{m_{n2}y}{\Omega_{n2}}}dy=\frac{\gamma \left(m_{n2},ym_{n2}/\Omega_{n2}\right)}{\Gamma (m_{n2})}$$
where $\Gamma (z)=\int_{0}^{\infty }e^{-t}t^{z-1}dt$ is the Gamma function ([26], Equation (8.310.1)). When z takes integer values, $\Gamma (z)=\left(z-1\right)!$ and $\gamma (z,x)=n!\left[1-e^{-x}\left(\sum_{m=0}^{z}\frac{x^{m}}{m!}\right)\right]$ ([26], Equation (8.339.1), Equation (8.352.1)). Therefore, $F_{{\left|h_{nkm}^{}\right|}^{2}}(y)$ and $F_{{\left|g_{nkl}^{}\right|}^{2}}(y)$ can be rewritten as
(13)$$F_{{\left|h_{nkm}^{}\right|}^{2}}(y)=1-e^{-\frac{ym_{n1}}{\Omega_{n1}}}\sum_{p=0}^{m_{n1}-1}\frac{{\left(ym_{n1}/\Omega_{n1}\right)}^{p}}{p!}$$
(14)$$F_{{\left|g_{nkl}^{}\right|}^{2}}(y)=1-e^{-\frac{ym_{n2}}{\Omega_{n2}}}\sum_{r=0}^{m_{2}-1}\frac{{\left(ym_{n2}/\Omega_{n2}\right)}^{r}}{r!}$$

By consulting the higher order statistics, the CDF of ${\left|h_{nkm}^{\max}\right|}^{2}$ and ${\left|g_{nkl}^{\max}\right|}^{2}$ can be obtained as
(15)$$F_{{\left|h_{{}_{nkm}}^{\max}\right|}^{2}}(y)={\left(1-e^{-uy}\sum_{p=0}^{m_{n1}-1}\frac{{\left(uy\right)}^{p}}{p!}\right)}^{M}$$
(16)$$F_{{\left|g_{{}_{nkl}}^{\max}\right|}^{2}}(y)={\left(1-e^{-\nu y}\sum_{r=0}^{m_{n}{}_{2}-1}\frac{{\left(\nu y\right)}^{r}}{r!}\right)}^{L}$$
where $u=m_{n1}/\Omega_{n1}$, $v=m_{n2}/\Omega_{n2}$.Taking the derivative of the above formulas, the PDF of ${\left|h_{nkm}^{\max}\right|}^{2}$ and ${\left|g_{nkl}^{\max}\right|}^{2}$ can be expressed as follows:(17)$$f_{{\left|h_{{}_{nkm}}^{\max}\right|}^{2}}(y)=\frac{Mu^{m_{n1}}y^{m_{n1}-1}e^{-uy}}{\left(m_{n1}-1\right)!}{\left(1-e^{-uy}\sum_{p=0}^{m_{n1}-1}\frac{{\left(uy\right)}^{p}}{p!}\right)}^{M-1}$$
(18)$$f_{{\left|g_{{}_{nkl}}^{\max}\right|}^{2}}(y)=\frac{L\nu^{m_{n2}}y^{m_{n2}-1}e^{-vy}}{\left(m_{n2}-1\right)!}{\left(1-e^{-\nu y}\sum_{r=0}^{m_{n}{}_{2}-1}\frac{{\left(\nu y\right)}^{r}}{r!}\right)}^{L-1}$$

## 3. Performance Analysis

### 3.1. Schematic Design

We assume that the communication requirement of the system model is to maximize the transmission rate of User 2 when the communication transmission of User 1 is not interrupted. To solve the problem of excessive computational complexity after combining multi-relay selection and multi-antenna selection in this model, this paper proposes a joint relay-and-antenna selection scheme, which can greatly reduce the amount of system computation. The specific schemes are as follows:

(1) Select a subset $S_{1}$ of relay nodes that enables the signal $x_{1}$ and the signal $x_{2}$ to be decoded normally. It can be expressed as
(19)$$S_{1}=\left\{{\left|i_{n}\right|}^{2}>\eta ,n\in N\right\}$$
where $\eta =\max(\frac{\varepsilon_{1}}{\left(\gamma_{1}-\gamma_{2}\varepsilon_{1}\right)\rho_{sn}},\frac{\varepsilon_{2}}{\gamma_{2}\rho_{sn}})$, $\varepsilon_{1}=2^{2R_{1}}-1$, $\varepsilon_{2}=2^{2R_{2}}-1$, $R_{1}$ and $R_{2}$ are the target rates of User 1 and User 2, respectively.

(2) Select the maximum values $h_{nkm}^{\max}$ and $g_{nkl}^{\max}$ from the channel vectors $h_{nk}$ and $g_{nk}$ to form a subset $S_{2}=\left\{(h_{nkm}^{\max},g_{nkl}^{\max}),n\in S_{1},k\in K\right\}$.
(20)$$h_{nkm}^{\max}=\max(h_{nk1},h_{nk2},\ldots ,h_{nkM})$$
(21)$$g_{nkl}^{\max}=\max(g_{nk1},g_{nk2},\ldots ,g_{nkL})$$

(3) Select a subset $S_{3}$ from subset $S_{2}$ that enables the signal $x_{1}$ to successfully decode at User 1 and User 2 with $q_{nk}=\min({\left|h_{nkm}^{\max}\right|}^{2},{\left|g_{nkl}^{\max}\right|}^{2})$. Subset $S_{3}$ can be expressed as
(22)$$S_{3}=\left\{\min({\left|h_{nkm}^{\max}\right|}^{2},{\left|g_{nkl}^{\max}\right|}^{2})>\frac{\varepsilon_{1}}{\left(\gamma_{1}-\gamma_{2}\varepsilon_{1}\right)\rho_{nd}},n\in S_{1},k\in S_{2}\right\}$$

(4) Select the maximum value $q_{n}$ of $q_{nk}$ corresponding to relay n to form subset $S_{4}$. Subset $S_{4}$ can be expressed as
(23)$$S_{4}=\left\{q_{n}=\max(q_{n1},q_{n2},\ldots ,q_{nk}),n\in S_{3},k\in S_{3}\right\}$$

(5) Select the largest $q_{n}(n\in S_{4})$ from subset $S_{4}$ to maximize the User 2 rate:(24)$$\left\{n^{*},k^{*},m^{*},l^{*}\right\}=\arg\max\left\{\gamma_{2}\rho_{n2}{\left|g_{nkl}\right|}^{2},n\in S_{4},k\in S_{4}\right\}$$
where $n^{*}$ represents the best relay node. $k^{*},m^{*},l^{*}$ respectively represent the antennas corresponding to relay $n^{*}$, User 1 and User 2 under the optimal selection condition.

### 3.2. The Exact Outage Probability Expressions for NOMA Users

(1) User 1: Define $A_{1}$ as an event: The signal $x_{1}$ can be successfully decoded at relay n. Define $A_{2}$ as an event: User 1 can successfully decode the signal $x_{1}$. The probability that any relay node n cannot successfully decode the signal $x_{1}$ can be expressed as
(25)$$P({\bar{A}}_{1})=\mathrm{Pr}({\left|i_{n}\right|}^{2}<\frac{\varepsilon_{1}}{\left(\gamma_{1}-\gamma_{2}\varepsilon_{1}\right)\rho_{nd}})=F_{{\left|i_{n}\right|}^{2}}(\frac{\varepsilon_{1}}{\left(\gamma_{1}-\gamma_{2}\varepsilon_{1}\right)\rho_{nd}})$$

The probability that the signal forwarded by relay node n is interrupted at User 1 can be expressed as:(26)$$P({\bar{A}}_{2})={\left(\mathrm{Pr}({\left|h_{nkm}^{\max}\right|}^{2}<\frac{\varepsilon_{1}}{\left(\gamma_{1}-\gamma_{2}\varepsilon_{1}\right)\rho_{nd}})\right)}^{K}$$

Therefore, the outage probability of User 1 can be expressed as
(27)$$\begin{array}{ll} P(\bar{A}) & ={\sum_{n=1}^{N}\left(\begin{array}{c} N\\ n \end{array}\right)\left(P({\bar{A}}_{2})\right)}^{n}(1-P({\bar{A}}_{1}){)}^{n}(P({\bar{A}}_{1}){)}^{N-n}+(P({\bar{A}}_{1}){)}^{N}\\ & =\sum_{n=0}^{N}\left(\begin{array}{c} N\\ n \end{array}\right){\left(F_{{\left|h_{nkm}^{\max}\right|}^{2}}(\frac{\varepsilon_{1}}{\left(\gamma_{1}-\gamma_{2}\varepsilon_{1}\right)\rho_{nd}})\right)}^{Kn}(1-F_{{\left|i_{n}\right|}^{2}}(\frac{\varepsilon_{1}}{\left(\gamma_{1}-\gamma_{2}\varepsilon_{1}\right)\rho_{nd}}){)}^{n}(F_{{\left|i_{n}\right|}^{2}}(\frac{\varepsilon_{1}}{\left(\gamma_{1}-\gamma_{2}\varepsilon_{1}\right)\rho_{nd}}){)}^{N-n} \end{array}$$

(2) User 2: Define $B_{1}$ as an event: any relay n can successfully decode the signal $x_{1}$ and the signal $x_{2}$. Define $B_{2}$ as an event: the superimposed signal forwarded by the *k*–th antenna of any relay n can successfully decode the signal $x_{1}$ at User 1 and User 2. Define $B_{3}$ as an event: the superimposed signal forwarded through any relay n can successfully decode the signal $x_{2}$ at User 2.

In the first time slot, the probability that all relay nodes cannot decode the signal $x_{1}$ and the signal $x_{2}$ can be expressed as
(28)$$P({\bar{B}}_{1})=\mathrm{Pr}(\left|S_{1}\right|=0)=\overset{N}{\underset{n=1}{\Pi }}\mathrm{Pr}({\left|i_{n}\right|}^{2}<\eta )={\left(F_{{\left|i_{n}\right|}^{2}}(\eta )\right)}^{N}$$

The probability that at least one user cannot decode the signal $x_{1}$ from the superimposed signal can be expressed as
(29)$$\begin{array}{ll} P({\bar{B}}_{2}) & =\mathrm{Pr}\left(\left|S_{3}\right|=0\right)=\prod_{k=1}^{K}\mathrm{Pr}\left(\min\left(h_{nk}^{\max},g_{nk}^{\max}\right)<\frac{\varepsilon_{1}}{\left(\gamma_{1}-\gamma_{2}\varepsilon_{1}\right)\rho_{nd}}\right)\\ & =\prod_{k=1}^{K}\mathrm{Pr}\left(\gamma_{2}\leq 0\right)=\prod_{k=1}^{K}\mathrm{Pr}\left(q_{nk}\leq \frac{\varepsilon_{1}}{\rho_{nd}}\right)={\left[1-\left(1-F_{{\left|h_{nkm}\right|}^{2}}(\frac{\varepsilon_{1}}{\rho_{nd}})\right)\left(1-F_{{\left|g_{nkl}\right|}^{2}}(\frac{\varepsilon_{1}}{\rho_{nd}})\right)\right]}^{K} \end{array}$$

On the basis that the relay nodes can successfully decode and forward the superimposed signal, and the users can decode the signal $x_{1}$ normally. The probability that User 2 cannot decode the signal $x_{2}$ can be derived as
(30)$$\begin{array}{ll} P(B_{1}B_{2}{\bar{B}}_{3}) & =\mathrm{Pr}(\gamma_{2}\rho_{n2}{\left|g_{{}_{nkl}}^{*}\right|}^{2}<\varepsilon_{2},\left|S_{3}\right|>0,\left|S_{1}\right|>0)\\ & =\sum_{n=1}^{N}\mathrm{Pr}(\gamma_{2}\rho_{n2}{\left|g_{{}_{nkl}}^{*}\right|}^{2}<\varepsilon_{2},\left|S_{3}\right|>0|\left|S_{1}\right|=n)\mathrm{Pr}(\left|S_{1}\right|=n)\\ & ={\sum_{n=1}^{N}\left(\mathrm{Pr}(\gamma_{2}\rho_{n2}{\left|g_{{}_{nkl}}^{k\max}\right|}^{2}<\varepsilon_{2},\left|S_{3}\right|>0|\left|S_{1}\right|=n)\right)}^{n}\mathrm{Pr}(\left|S_{1}\right|=n) \end{array}$$
where:(31)$$\begin{array}{l} \mathrm{Pr}(\gamma_{2}\rho_{n2}{\left|g_{{}_{nkl}}^{k\max}\right|}^{2}<\varepsilon_{2},\left|S_{3}\right|>0)\\ =\sum_{k=1}^{K}\left(\mathrm{Pr}(\gamma_{2}\rho_{n2}{\left|g_{{}_{nkl}}^{k\max}\right|}^{2}<\varepsilon_{2}|\left|S_{3}\right|=k)\right)\mathrm{Pr}(\left|S_{3}\right|=k)\\ ={\sum_{k=1}^{K}\left(\mathrm{Pr}(\gamma_{2}\rho_{n2}{\left|g_{{}_{nkl}}^{\max}\right|}^{2}<\varepsilon_{2}|\left|S_{3}\right|=k)\right)}^{k}\mathrm{Pr}(\left|S_{3}\right|=k) \end{array}$$

For any relay n in subset $\left|S_{1}\right|>0$, combining the two expressions of $SINR_{1}^{1}\geq \varepsilon_{1}$ and $SINR_{2}^{1}\geq \varepsilon_{1}$, the range of $\gamma_{2}$ when the signal $x_{1}$ can be successfully decoded is
(32)$$\gamma_{2}=\min\left\{\frac{\rho_{nd}{\left|h_{nkm}\right|}^{2}-\varepsilon_{1}}{\rho_{nd}{\left|h_{nkm}\right|}^{2}\left(1+\varepsilon_{1}\right)},\right.\left.\max\left\{0,\frac{\rho_{nd}{\left|g_{nkl}\right|}^{2}-\varepsilon_{1}}{\rho_{nd}{\left|g_{nkl}\right|}^{2}\left(1+\varepsilon_{1}\right)}\right\}\right\}$$

Bring the above formula into expression $q_{nk}=\min({\left|h_{nkm}^{\max}\right|}^{2},{\left|g_{nkl}^{\max}\right|}^{2})$, $\gamma_{2}$ can be rewritten as $\gamma_{2}=\frac{\rho_{nd}q_{nk}-\varepsilon_{1}}{\rho_{nd}q_{nk}\left(1+\varepsilon_{1}\right)}$. In the set of elements that satisfy $\left|S_{3}\right|>0$, the probability that User 2 cannot successfully decode the signal $x_{2}$ can be expressed as

(33)$$\begin{array}{l} \mathrm{Pr}(\gamma_{2}\rho_{n2}{\left|g_{{}_{nkl}}^{\max}\right|}^{2}<\varepsilon_{2}|k\in S_{3},\left|S_{3}\right|>0)\\ =\mathrm{Pr}(\frac{\rho_{nd}q_{nk}-\varepsilon_{1}}{\rho_{nd}q_{nk}\left(1+\varepsilon_{1}\right)}<\frac{\varepsilon_{2}}{\rho_{n2}{\left|g_{{}_{nkl}}^{\max}\right|}^{2}}|k\in S_{3},\left|S_{3}\right|>0)\\ =\mathrm{Pr}(\frac{\rho_{nd}q_{nk}-\varepsilon_{1}}{q_{nk}\left(1+\varepsilon_{1}\right)}<\frac{\varepsilon_{2}}{{\left|g_{{}_{nkl}}^{\max}\right|}^{2}}|q_{nk}>\frac{\varepsilon_{1}}{\rho_{nd}})\\ =\frac{\mathrm{Pr}(\frac{\rho_{nd}{\left|h_{nkm}^{\max}\right|}^{2}-\varepsilon_{1}}{{\left|h_{nkm}^{\max}\right|}^{2}\left(1+\varepsilon_{1}\right)}<\frac{\varepsilon_{2}}{{\left|g_{{}_{nkl}}^{\max}\right|}^{2}},{\left|g_{{}_{nkl}}^{\max}\right|}^{2}\geq {\left|h_{nkm}^{\max}\right|}^{2}>\frac{\varepsilon_{1}}{\rho_{nd}})}{\mathrm{Pr}(q_{nk}>\frac{\varepsilon_{1}}{\rho_{nd}})}+\frac{\mathrm{Pr}(\frac{\rho_{nd}{\left|g_{{}_{nkl}}^{\max}\right|}^{2}-\varepsilon_{1}}{{\left|g_{{}_{nkl}}^{\max}\right|}^{2}\left(1+\varepsilon_{1}\right)}<\frac{\varepsilon_{2}}{{\left|g_{{}_{nkl}}^{\max}\right|}^{2}},{\left|h_{nkm}^{\max}\right|}^{2}>{\left|g_{{}_{nkl}}^{\max}\right|}^{2}>\frac{\varepsilon_{1}}{\rho_{nd}})}{\mathrm{Pr}(q_{nk}>\frac{\varepsilon_{1}}{\rho_{nd}})} \end{array}$$

Let the molecules of the above formula be $J_{1}$ and $J_{2}$, respectively. $J_{1}$ can be expressed as

(34)$$\begin{array}{ll} J_{1} & =\mathrm{Pr}(\frac{\rho_{nd}{\left|h_{nkm}^{\max}\right|}^{2}-\varepsilon_{1}}{{\left|h_{nkm}^{\max}\right|}^{2}\left(1+\varepsilon_{1}\right)}<\frac{\varepsilon_{2}}{{\left|g_{{}_{nkl}}^{\max}\right|}^{2}},{\left|g_{{}_{nkl}}^{\max}\right|}^{2}\geq {\left|h_{nkm}^{\max}\right|}^{2}>\frac{\varepsilon_{1}}{\rho_{nd}})\\ & =\mathrm{Pr}(\frac{\varepsilon_{2}{\left|h_{nkm}^{\max}\right|}^{2}\left(1+\varepsilon_{1}\right)}{\rho_{nd}{\left|h_{nkm}^{\max}\right|}^{2}-\varepsilon_{1}}>{\left|g_{{}_{nkl}}^{\max}\right|}^{2}\geq {\left|h_{nkm}^{\max}\right|}^{2}>\frac{\varepsilon_{1}}{\rho_{nd}})\\ & =\int_{\frac{\varepsilon_{1}}{\rho_{nd}}}^{\frac{a_{2}}{\rho_{nd}}}f_{{\left|h_{nkm}\right|}^{2}}(x)dx\int_{x}^{\frac{a_{1}x}{\rho_{nd}x-\varepsilon_{1}}}f_{{\left|g_{nkl}\right|}^{2}}(y)dy\\ & =\underset{\Psi_{1}}{\underset{︸}{\int_{\frac{\varepsilon_{1}}{\rho_{nd}}}^{\frac{a_{2}}{\rho_{nd}}}f_{{\left|h_{nkm}\right|}^{2}}(x)F_{{\left|g_{nkl}\right|}^{2}}(\frac{a_{1}x}{\rho_{nd}x-\varepsilon_{1}})dx}}-\underset{\Psi_{2}}{\underset{︸}{\int_{\frac{\varepsilon_{1}}{\rho_{nd}}}^{\frac{a_{2}}{\rho_{nd}}}f_{{\left|h_{nkm}\right|}^{2}}(x)F_{{\left|g_{nkl}\right|}^{2}}(x)dx}} \end{array}$$

Expand and integrate $\Psi_{1}$ and $\Psi_{2}$ through the polynomial theorem. We can get the expressions (35) and (36):

(35)$$\begin{array}{ll} \Psi_{1}= & \sum_{i=0}^{M-1}\left(\begin{array}{c} M-1\\ i \end{array}\right)\frac{\Xi_{i}\Phi_{i}Mu^{m_{n1}}}{\left(m_{n1}-1\right)!}\sum_{j=0}^{L}\left(\begin{array}{l} L\\ j \end{array}\right){\left(-1\right)}^{i+j}\Xi_{j}\Phi_{j}e^{\frac{-u\varepsilon_{1}(i+1)-jva_{1}}{\rho_{nd}}}\\ & \times \sum_{w=0}^{{{}^{m_{n1}-1}}^{+\tilde{i}}}\left(\begin{array}{c} {{}^{m_{n1}-1}}^{+\tilde{i}}\\ w \end{array}\right){\left(\frac{\varepsilon_{1}}{\rho_{nd}}\right)}^{m_{n1}-1}{}^{+\tilde{i}-w+\tilde{j}-z}{\left(\frac{a_{1}}{\rho_{nd}}\right)}^{\tilde{j}}\sum_{z=0}^{\tilde{j}}\left(\begin{array}{l} {}^{\tilde{j}}\\ z \end{array}\right)\int_{0}^{\frac{a_{1}}{\rho_{nd}}}e^{-ut(i+1)}t^{w+z-\tilde{j}}e^{\frac{-jva_{1}\varepsilon_{1}}{t\rho_{{}_{nd}}^{2}}}dt \end{array}$$

(36)$$\begin{array}{ll} \Psi_{2}= & \sum_{i=0}^{M-1}\left(\begin{array}{c} M-1\\ i \end{array}\right)\frac{\Xi_{i}\Phi_{i}Mu^{m_{n1}}}{\left(m_{n1}-1\right)!}\sum_{j=0}^{L}\left(\begin{array}{c} L\\ j \end{array}\right)(-1{)}^{i+j}\Xi_{j}\Phi_{j}e^{-\frac{\varepsilon_{1}}{\rho_{nd}}(ui+u+jv)}\\ & \times \sum_{r=0}^{{{}^{m_{n1}-1}}^{+\tilde{i}+\tilde{j}}}\left(\begin{array}{c} {{}^{m_{n1}-1}}^{+\tilde{i}+\tilde{j}}\\ r \end{array}\right){\left(\frac{\varepsilon_{1}}{\rho_{nd}}\right)}^{m_{n1}-1}{}^{+\tilde{i}+\tilde{j}-r}(ui+u+jv{)}^{-r-1}\gamma (r+1,\frac{a_{1}}{\rho_{nd}}(ui+u+jv)) \end{array}$$

Similarly, $J_{2}$ can be expressed as (37)$$\begin{array}{ll} J_{2} & =\mathrm{Pr}(\frac{\rho_{nd}{\left|g_{{}_{nkl}}^{\max}\right|}^{2}-\varepsilon_{1}}{{\left|g_{{}_{nkl}}^{\max}\right|}^{2}\left(1+\varepsilon_{1}\right)}<\frac{\varepsilon_{2}}{{\left|g_{{}_{nkl}}^{\max}\right|}^{2}},{\left|h_{nkm}^{\max}\right|}^{2}>{\left|g_{{}_{nkl}}^{\max}\right|}^{2}>\frac{\varepsilon_{1}}{\rho_{nd}})\\ & =\mathrm{Pr}({\left|g_{{}_{nkl}}^{\max}\right|}^{2}<\frac{\varepsilon_{2}\varepsilon_{1}+\varepsilon_{1}+\varepsilon_{2}}{\rho_{nd}},{\left|h_{nkm}^{\max}\right|}^{2}>{\left|g_{{}_{nkl}}^{\max}\right|}^{2}>\frac{\varepsilon_{1}}{\rho_{nd}})\\ & =\int_{\frac{\varepsilon_{1}}{\rho_{nd}}}^{\frac{a_{2}}{\rho_{nd}}}f_{{\left|g_{nkl}\right|}^{2}}(y)dy\int_{y}^{\infty }f_{{\left|h_{nkm}\right|}^{2}}(x)dx=\underset{\Psi_{3}}{\underset{︸}{\int_{\frac{\varepsilon_{1}}{\rho_{nd}}}^{\frac{a_{2}}{\rho_{nd}}}f_{{\left|g_{nkl}\right|}^{2}}(y)dy}}-\underset{\Psi_{4}}{\underset{︸}{\int_{\frac{\varepsilon_{1}}{\rho_{nd}}}^{\frac{a_{2}}{\rho_{nd}}}f_{{\left|g_{nkl}\right|}^{2}}(y)F_{{\left|h_{nkm}\right|}^{2}}(y)dy}} \end{array}$$

Expand and integrate $\Psi_{3}$ and $\Psi_{4}$ through the polynomial theorem. We can get the expressions (38) and (39):(38)$$\Psi_{3}=\sum_{j=0}^{L}\left(\begin{array}{c} L\\ j \end{array}\right){\left(-1\right)}^{j}\Xi_{j}\Phi_{j}\left[{\left(\frac{a_{2}}{\rho_{nd}}\right)}^{\tilde{j}}e^{\frac{-jva_{2}}{\rho_{nd}}}-{\left(\frac{\varepsilon_{1}}{\rho_{nd}}\right)}^{\tilde{j}}e^{\frac{-jv\varepsilon_{1}}{\rho_{nd}}}\right]$$
(39)$$\begin{array}{ll} \Psi_{4}= & \sum_{j=0}^{L-1}\left(\begin{array}{c} L-1\\ j \end{array}\right)\frac{\Xi_{j}\Phi_{j}Lv^{m_{n2}}}{\left(m_{n2}-1\right)!}\sum_{i=0}^{M}\left(\begin{array}{c} M\\ i \end{array}\right)(-1{)}^{i+j}\Xi_{i}\Phi_{i}e^{-\frac{\varepsilon_{1}}{\rho_{nd}}(ui+u+jv)}\\ & \times \sum_{r=0}^{{{}^{m_{n2}-1}}^{+\tilde{i}+\tilde{j}}}\left(\begin{array}{c} {{}^{m_{n2}-1}}^{+\tilde{i}+\tilde{j}}\\ r \end{array}\right){\left(\frac{\varepsilon_{1}}{\rho_{nd}}\right)}^{m_{n2}-1}{}^{+\tilde{i}+\tilde{j}-r}(ui+u+jv{)}^{-r-1}\gamma (r+1,\frac{a_{1}}{\rho_{nd}}(ui+u+jv)) \end{array}$$
where:(40)$$\begin{array}{l} a=\eta +\varepsilon_{1},a_{1}=\varepsilon_{2}(\varepsilon_{1}+1),a_{2}=a_{1}+\varepsilon_{1},i=i_{0}+i_{1}+\cdot \cdot \cdot +i_{m_{n1}-1},j=j_{0}+j_{1}+\cdot \cdot \cdot +j_{m_{n2}-1}\\ \Xi_{i}=\sum_{i_{1}=0}^{i}\sum_{i_{2}=0}^{i-i_{1}}\cdots \sum_{i_{m_{n1}-1}=0}^{i-i_{1}-\cdots -i_{m_{n1}-2}},\Xi_{j}=\sum_{j_{1}=0}^{j}\sum_{j_{2}=0}^{j-j_{1}}\cdots \sum_{j_{m_{n2}-1}=0}^{j-j_{1}-\cdots -j_{m_{n2}-2}},\\ \Phi_{i}=\left(\begin{array}{c} i\\ i_{1} \end{array}\right)\left(\begin{array}{c} i-i_{1}\\ i_{2} \end{array}\right)\cdots \left(\begin{array}{c} i-i_{1}-i_{2}-\cdots -i_{m_{n1}-2}\\ i_{m_{n1}-1} \end{array}\right){\prod_{p=0}^{m_{n1}-1}\left(\frac{u^{p}}{p!}\right)}^{i_{p}},\tilde{i}=0*i_{0}+1*i_{1}+\ldots +(m_{n1}-1)*i_{m_{n1}-1}\\ \Phi_{j}=\left(\begin{array}{c} j\\ j_{1} \end{array}\right)\left(\begin{array}{c} j-j_{1}\\ j_{2} \end{array}\right)\cdots \left(\begin{array}{c} j-j_{1}-j_{2}-\cdots -j_{m_{n2}-2}\\ j_{m_{n2}-1} \end{array}\right){\prod_{s=0}^{m_{n2}-1}\left(\frac{v^{s}}{s!}\right)}^{j_{s}},\tilde{j}=0*j_{0}+1*j_{1}+\ldots +(m_{n2}-1)*j_{m_{n2}-1} \end{array}$$

Comprehensive expressions (28)–(40), the exact expression of the outage probability for User 2 can be obtained as the expression (41):(41)$$P(\bar{B})={\sum_{n=0}^{N}\left(\begin{array}{c} N\\ n \end{array}\right)\left({\sum_{k=0}^{K}\left(\begin{array}{c} K\\ k \end{array}\right){\left(\Psi_{1}-\Psi_{2}+\Psi_{3}-\Psi_{4}\right)}^{k}\left(1-\left(1-F_{{\left|h_{nkm}\right|}^{2}}(\frac{\varepsilon_{1}}{\rho_{nd}})\right)\left(1-F_{{\left|g_{nkl}\right|}^{2}}(\frac{\varepsilon_{1}}{\rho_{nd}})\right)\right)}^{K-k}\right)}^{n}{\left(1-F_{{\left|i_{n}\right|}^{2}}(\eta )\right)}^{n}{\left(F_{{\left|i_{n}\right|}^{2}}(\eta )\right)}^{N-n}$$

### 3.3. The Asymptotic Outage Probability Expressions for NOMA Users

When $\rho_{sn}\to \infty$, the asymptotic CDF expression of the communication link from the satellite to the relay n can be simplified to
(42)$$F_{{\left|i_{n}\right|}^{2}}(x)=\alpha_{{\left|i_{n}\right|}^{2}}\sum_{k=0}^{\infty }\frac{{\left(m_{{\left|i_{n}\right|}^{2}}\right)}_{k}\delta_{{\left|i_{n}\right|}^{2}}^{k}}{{\left(k!\right)}^{2}\beta_{{\left|i_{n}\right|}^{2}}^{k+1}}\gamma \left(k+1,\beta_{{\left|i_{n}\right|}^{2}}x\right)\approx \alpha_{{\left|i_{n}\right|}^{2}}x$$

When $\rho_{nd}\to \infty$, the asymptotic CDF expression of the communication link from the relay n to the users can be simplified to
(43)$$F_{{\left|h_{{}_{nkm}}^{\max}\right|}^{2}}(y)={\left(1-e^{-uy}\sum_{p=0}^{m_{n1}-1}\frac{{\left(uy\right)}^{p}}{p!}\right)}^{M}\approx {\left(\frac{{\left(uy\right)}^{m_{n1}}}{m_{n1}!}\right)}^{M}$$
(44)$$F_{{\left|g_{{}_{nkl}}^{\max}\right|}^{2}}(y)={\left(1-e^{-\nu y}\sum_{r=0}^{m_{n}{}_{2}-1}\frac{{\left(\nu y\right)}^{r}}{r!}\right)}^{L}\approx {\left(\frac{{\left(vy\right)}^{m_{n2}}}{m_{n2}!}\right)}^{L}$$

The corresponding asymptotic PDF expression can be simplified to
(45)$$f_{{\left|h_{{}_{nkm}}^{\max}\right|}^{2}}(y)\approx \frac{Mu^{Mm_{n1}}y^{Mm_{n1}-1}(1-uy)}{\left(m_{n1}-1\right)!{\left(m_{n1}!\right)}^{M-1}}$$
(46)$$f_{{\left|g_{{}_{nkl}}^{\max}\right|}^{2}}(y)\approx \frac{L\nu^{Lm_{n2}}y^{Lm_{n2}-1}(1-vy)}{\left(m_{n2}-1\right)!{\left(m_{n1}!\right)}^{L-1}}$$

Substituting the expressions (42) and (43) into the expression (27), the expression of the asymptotic outage probability of User 1 can be written as the expression (48):(47)$$P(\bar{A})\approx \sum_{n=0}^{N}\left(\begin{array}{c} N\\ n \end{array}\right){\left(\frac{u^{Mm_{n1}}\varepsilon_{1}{}^{Mm_{n1}}}{{\left(m_{n1}!\right)}^{M}\rho_{nd}{}^{M}{\left(\gamma_{1}-\gamma_{2}\varepsilon_{1}\right)}^{M}}\right)}^{Kn}{\left(1-\frac{\alpha_{{\left|i_{n}\right|}^{2}}\varepsilon_{1}}{\left(\gamma_{1}-\gamma_{2}\varepsilon_{1}\right)\rho_{nd}}\right)}^{n}{\left(\frac{\alpha_{{\left|i_{n}\right|}^{2}}\varepsilon_{1}}{\left(\gamma_{1}-\gamma_{2}\varepsilon_{1}\right)\rho_{nd}}\right)}^{N-n}$$

Substituting expressions (42)–(46) into expressions (35)–(41), the expression of the asymptotic outage probability of User 2 can be obtained as the expression (48):(48)$$P(\bar{B})\approx {\sum_{n=0}^{N}\left(\begin{array}{c} N\\ n \end{array}\right)\left({\sum_{k=0}^{K}\left(\begin{array}{c} K\\ k \end{array}\right){\left(\Psi_{1}-\Psi_{2}+\Psi_{3}-\Psi_{4}\right)}^{k}\left(1-\left(1-\frac{u^{Mm_{n1}}\varepsilon_{1}{}^{Mm_{n1}}}{{\left(m_{n1}!\right)}^{M}\rho_{nd}{}^{M}}\right)\left(1-\frac{v^{Lm_{n2}}\varepsilon_{1}{}^{Lm_{n2}}}{{\left(m_{n2}!\right)}^{L}\rho_{nd}{}^{L}}\right)\right)}^{K-k}\right)}^{n}{\left(1-\alpha_{{\left|i_{n}\right|}^{2}}\eta \right)}^{n}{\left(\alpha_{{\left|i_{n}\right|}^{2}}\eta \right)}^{N-n}$$

## 4. Numeric Results

By using numeric simulation, we analyze and compare the impact of the number of relays and antenna configuration on the NOMA-based HSTRN outage performance and prove the superiority of the proposed joint relay-and-antenna selection scheme. In the system simulation, we assume that the communication link from satellite to terrestrial relay undergoes heavy shadowing (HS) with $\left(m_{sn},b_{sn},\Omega_{sn}\right)$ = (0.739, 0.063, 8.97 × 10^−4^) or average shadowing (AS) with $\left(m_{sn},b_{sn},\Omega_{sn}\right)$ = (10.1, 0.126, 0.835) or light shadowing (AS) with $\left(m_{sn},b_{sn},\Omega_{sn}\right)$ = (19.4, 0.158, 1.29), $\rho_{sn}=\rho_{nd}$[27]. In order to distinguish the difference between channels of different users, the parameter between the user group and the relay are set to $m_{n1}=0.8$, $\Omega_{n1}=0.8$, $m_{n2}=1$, $\Omega_{n2}=1$ [28,29].

Figure 2 plots the outage probability for NOMA and TDMA under different shadowed Rician fading. Observing Figure 2, we can conclude that compared to TDMA, the proposed scheme can realize better outage probability. As the communication link transitions from HS to LS, the outage performance of User 1 and User 2 continues to improve. We can also find that the increase in outage probability between HS and AS is much greater than between AS and LS. The main reason for this phenomenon is that $\Omega_{{\left|i_{n}\right|}^{2}}$ has increased by 930 times and 2 times, respectively. The substantial increase in the received power significantly improves outage performance.

Figure 3 depicts the impact of the number of relays and terrestrial nodes antenna configuration on users’ outage probability. It can be observed that the proposed scheme has better outage performance than TDMA. Regardless of the increase in the number of relays or the number of terrestrial nodes antennas, users’ outage performance can be improved. However, the outage performance of users is more susceptible to the number of relays. The main reason is that the shadowed Rician fading from the satellite to the ground is much larger than the terrestrial Nakagami-m fading, resulting in the change in the number of relays dominating the change in the number of antennas.

In Figure 4, we respectively show how the user’s target rate affects the outage probability of cooperative NOMA and cooperative OMA. It can be seen from the Figure that the outage probability of users based on NOMA is always better than that of TDMA. When the target rate of User 1 is small and fixed, its outage probability will not change with the change of the target rate of User 2. As the target rate of User 2 continues to increase, its outage performance under TDMA declines far greater than NOMA. The main reason is that the communication threshold of User 2 under TDMA increases with the increase in communication rate far greater than that of NOMA. At the same time, the system energy utilization rate of User 2 under TDMA is lower than that of NOMA.

In Table 1, we make a comparison between NOMA and TDMA schemes on the outage probability. It can be found that NOMA achieves better outage performance than that of TDMA. Through longitudinal comparison, the outage performance of users only improves 3 times when we set (N,K,M,L = 1,3,3,3). Under other conditions, the outage performance of users is increased by more than 10 times. Especially when R_1_ = 0.6 bit/s/Hz, the outage performance of User 2 is increased by 500 times. When the user adopts the TDMA scheme, each time slot can only serve one user. In order to complete the same transmission task per unit time, the user’s transmission rate in each time slot is doubled. The increase in the transmission rate leads to an increase in the corresponding threshold. Under the condition of total power limitation, the outage performance of users will decrease. At the same time, in the CR-NOMA solution, User 2 can make full use of the remaining power after ensuring that User 1 is not interrupted, which also greatly improves User 2’s outage performance.

## 5. Conclusions

In this paper, we proposed a NOMA-based joint relay-and-antenna selection scheme. By analyzing the SINR of each node in HSTRN under the DF strategy, the exact outage probability expression of each user and the asymptotic probability expression under high signal-to-noise ratio were derived. Through the use of software simulation, the effect of the number of relays and terrestrial nodes antenna configuration on the outage performance of the HSTRN system was studied. It proved the correctness of the proposed scheme and the superiority of the NOMA scheme compared to the TDMA scheme. At the same time, it also clarified the impact of each parameter configuration on system outage performance, which provided strong support for further research on HSTRN’s other strategies.

## Figures and Tables

**Figure 1 sensors-20-05177-f001:**
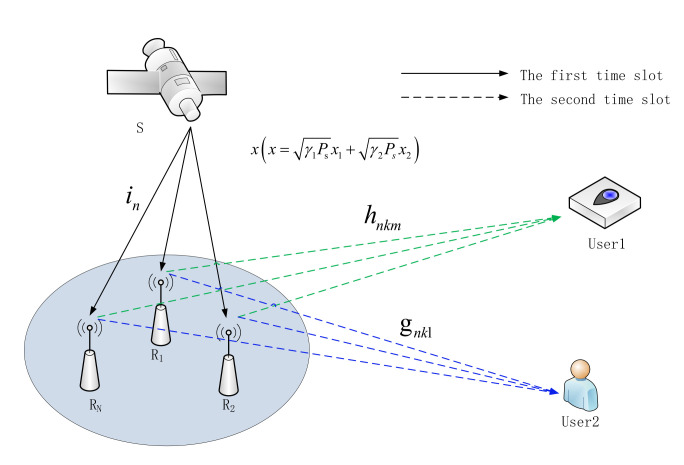
System model.

**Figure 2 sensors-20-05177-f002:**
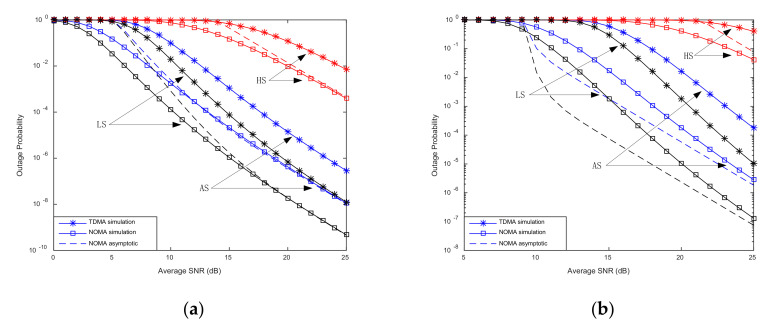
Outage probability vs. signal-to-noise ratio (SNR) for fading conditions. (**a**) User 1; (**b**) User 2.

**Figure 3 sensors-20-05177-f003:**
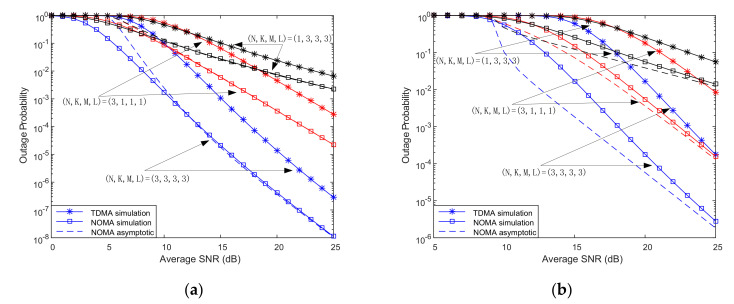
Outage probability vs. SNR for various relays and antenna numbers. (**a**) User1; (**b**) User2.

**Figure 4 sensors-20-05177-f004:**
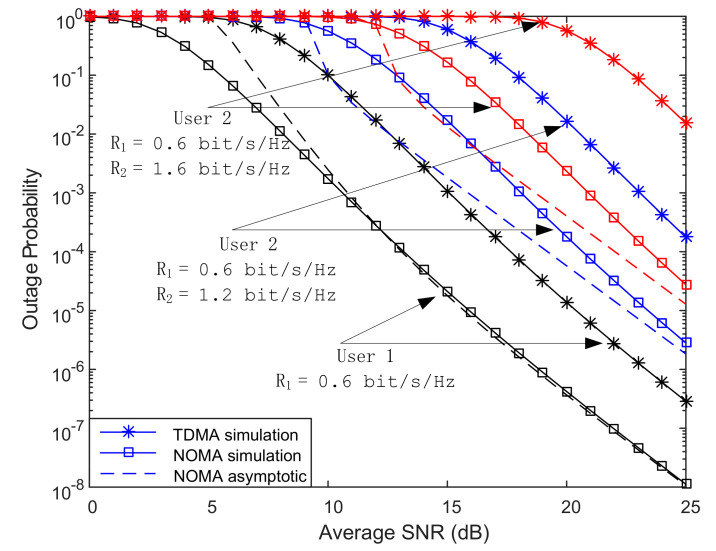
Outage probability vs. SNR for different target rates.

**Table 1 sensors-20-05177-t001:** Outage probability vs. various conditions.

		Condition	Terrestrial and Satellite Fading Conditions			Relay Numbers and Antenna Numbers			Target Rates	
	OP		HS	AS	LS	N,K,M,L = 1,3,3,3	N,K,M,L = 3,1,1,1	N,K,M,L = 3,3,3,3	R_1_ = 0.6 bit/s/Hz R_2_ = 1.2 bit/s/Hz	R_1_ = 0.6 bit/s/Hz R_2_ = 1.6 bit/s/Hz
User										
NOMA-User1			3.91 × 10^−4^	1.12 × 10^−8^	4.7 × 10^−10^	2.24 × 10^−3^	2.25 × 10^−5^	1.12 × 10^−8^	1.12 × 10^−8^	1.12 × 10^−8^
TDMA-User1			7.07 × 10^−3^	2.87 × 10^−7^	1.25 × 10^−8^	6.59 × 10^−3^	2.78 × 10^−4^	2.87 × 10^−7^	2.87 × 10^−7^	2.87 × 10^−7^
NOMA-User2			4.18 × 10^−2^	2.82 × 10^−6^	1.3 × 10^−7^	1.41 × 10^−2^	1.57 × 10^−4^	2.82 × 10^−6^	2.82 × 10^−6^	2.75 × 10^−5^
TDMA-User2			4.03 × 10^−1^	1.76 × 10^−4^	1.03 × 10^−5^	5.61 × 10^−2^	8.47 × 10^−3^	1.76 × 10^−4^	1.76 × 10^−4^	1.53 × 10^−2^

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
