# Peer review of "Performance Optimization of Hybrid Satellite-Terrestrial Relay Network Based on CR-NOMA"

_sensors, 2020, doi:10.3390/s20185177_

Round 1

Reviewer 1 Report

This work studied a joint relay-and-antenna selection scheme based on the cognitive radio scenario applied in the Hybrid Satellite-Terrestrial Relay Networks (HSTRN). It benefits to the maximum communication rate of the secondary user when the primary user maintains the best outage performance. They derived the exact outage probability formula of each user in the system and the asymptotic probability expression under high signal-to-noise ratio. The topic presented in this paper is timely, however, some weak pointed need be improved.

  1. Why the authors did not study more other antenna selection or relay selection to highlight which is the best fit to such system? Please elaborate this concern.
  2. It should be more reasonable if the authors compared your results with recent work, for example:

[R1] "NOMA in Cooperative Underlay Cognitive Radio Networks Under Imperfect SIC," IEEE Access, vol. 8, pp. 86180-86195, 2020.

[R2] "A Unified Framework for HS-UAV NOMA Networks: Performance Analysis and Location Optimization," IEEE Access, vol. 8, pp. 13329-13340, 2020.

  1. It is necessary to provide relevant references for eq. (10)-eq. (14)
  2. Lack of insight analysis and need more numerical results to show more advantages of such system.
  3. The presentation of the whole paper is till poor. English usage need be improved.

Author Response

Thanks for the careful review with detailed and professional advices on our work.Please find the attachment as the corresponding response of the first round review.

Reviewer 2 Report

Authors investigated the Hybrid Satellite-Terrestrial Relay using DF with NOMA scheme, however system model is simple. Before acceptance, some corrections are needed:

1. Need an English proofreading.
In 3.1 section heading. In reference 14, title spelling mistake.

2. In abstract, give some important results.

3. In introduction, give some reference papers of NOMA for satellite communication.

4. In results section, give a table comparison between TDMA and NOMA.

Author Response

(The authors gave the same response as above.)

Round 2

Reviewer 1 Report

I completely agree with response from authors. It can be published

Reviewer 2 Report

The authors have addressed my previous comments well in the revised version.